# Interactions of Fe–N–S Co-Doped Porous Carbons with Bacteria: Sorption Effect and Enzyme-Like Properties

**DOI:** 10.3390/ma13173707

**Published:** 2020-08-21

**Authors:** Andrzej Borkowski, Wojciech Kiciński, Mateusz Szala, Justyna Topolska, Paweł Działak, Marcin D. Syczewski

**Affiliations:** 1Faculty of Geology, Geophysics and Environmental Protection, AGH University of Science and Technology, Al. Mickiewicza 30, 30-059 Kraków, Poland; topolska@agh.edu.pl (J.T.); dzialak@agh.edu.pl (P.D.); 2Faculty of Advanced Technologies and Chemistry, Military University of Technology, Kaliskiego 2, 00-908 Warsaw, Poland; wojciech.kicinski@wat.edu.pl (W.K.); mateusz.szala@wat.edu.pl (M.S.); 3Faculty of Geology, University of Warsaw, Żwirki i Wigury 93, 02-089 Warsaw, Poland; marcinsyczewski@uw.edu.pl

**Keywords:** heteroatom-doped carbon, adhesion, bacteria, ROS, toxicity, Fe–N–C materials

## Abstract

Carbon-based (nano)materials doped with transition metals, nitrogen and other heteroatoms are considered active heterogeneous catalysts in a wide range of chemical processes. Recently they have been scrutinized as artificial enzymes since they can catalyze proton-coupled electron transfer reactions vital for living organisms. Herein, interactions between Gram-positive and Gram-negative bacteria and either metal-free N and/or S doped or metal containing Fe–N–S co-doped porous carbons are studied. The Fe- and N-co-doped porous carbons (Fe–N–C) exhibit enhanced affinity toward bacteria as they show the highest adsorption capacity. Fe–N–C materials also show the strongest influence on the bacteria viability with visible toxic effect. Both types of bacteria studied reacted to the presence of Fe-doped carbons in a similar manner, showing a decrease in dehydrogenases activity in comparison to controls. The N-coordinated iron-doped carbons (Fe–N–C) may exhibit oxidase/peroxidase-like activity and activate O_2_ dissolved in the solution and/or oxygen-containing species released by the bacteria (e.g., H_2_O_2_) to yield highly bactericidal reactive oxygen species. As Fe/N/ and/or S-doped carbon materials efficiently adsorb bacteria exhibiting simultaneously antibacterial properties, they can be applied, inter alia, as microbiological filters with enhanced biofouling resistance.

## 1. Introduction

Iron is one of the most important transition metals employed by metalloenzymes to facilitate a whole range of life-essential redox processes. In most metalloenzymes iron is bound to the macromolecules by nitrogen coordination (Fe–N_x_ moieties). Inorganic, iron-containing particles exhibit extraordinary, quasi-enzymatic activities, which were discovered during the study on the formation of reactive oxygen species (ROS) in the presence of Fe_3_O_4_ nanoparticles [1]. Due to this intriguing enzyme mimic activity, such materials were named nanozymes—nanomaterials with enzyme-like properties [2]. ROS may be generated abiotically when hydrogen peroxide reacts with Fe-containing nanomaterials leading to the production of superoxide radicals (^•^O_2_^−^), hydroxyl radicals (^•^OH), or singlet oxygen species (^1^O_2_). This process is similar to the activity of peroxidases, which catalyze the reaction of H_2_O_2_ with organic substrates via ROS generation [3,4,5,6]. For these reasons, the possible antibiotic properties of the nanozymes as well as their implementation for nanotechnology as the ROS determination agents have been studied intensively over the last several years. In case of carbon materials, numerous studies focus on bacteria sorption on activated carbons as the non-toxic material or on the toxicity of carbon nanotubes and graphene as the antibacterial material [7,8,9,10,11,12,13]. There are two main toxic mechanisms of carbon (nano)structures towards cells: (i) mechanical cell damage and (ii) reactive oxygen species generation leading to oxidative stress [14,15,16,17,18]. Considering the dimensions of heteroatom-doped carbon macro/micro-particles, their nano character is limited to the interior nano-porosity. Consequently, other factors play a pivotal role in the interactions between bacteria and doped carbon microstructures. Firstly, the highly developed surface of the porous, doped carbon particles yields a profusion of active surface sites that may interact with living cells. Secondly, the presence of heteroatoms (both metallic and non-metallic) within their structure can promote a variety of catalytic properties as well as ROS generation. The latter can occur due to the reaction of transition metal heteroatoms (e.g., Fe, Co, Cu, etc.) with oxygen leading to ^•^O_2_^−^ generation [19]. As a result, the Fe-doped carbons are able to mimic oxidase activity by producing H_2_O_2_ directly from atmospheric oxygen, yet simultaneously they can act as peroxidases catalyzing the hydrogen peroxide reduction [3,4,5,6]. Not only heteroatom-doped carbons, but a variety of Fe-containing materials and structures (e.g., F_2_O_3_, Fe_3_S_4_, etc.), exhibit such properties [20,21]. It is possible that the coexistence of oxidase/peroxidase-like activities is involved in the potential antibacterial activity of the Fe-doped carbon materials of various morphologies and dimensionalities (i.e., both nanostructures (such as nanofibers, nanotubes, etc.) and macroscopic objects such as porous macroparticles).

Summarizing, only a few studies investigating the possible antibacterial properties of Fe-doped carbon materials have been reported so far, and in most of them the addition of H_2_O_2_ is used as an external, artificial source of ROS simultaneously with the studied Fe-doped materials [4]. Simultaneous exposure of bacteria to Fe-doped materials and external (added) H_2_O_2_ precludes to indicate unambiguously the factors causing bacterial damage. Furthermore, the studied materials were often in the form of nanoparticles, and hence mechanical damages of cells could be the dominant reason of bactericidal properties. Consequently, the interesting question is whether Fe-doped carbon materials in macroscopic forms (macroparticles), without addition of any external ROS promoting compounds such as H_2_O_2_, also reveal significant antibacterial properties. Furthermore, unlike the antibacterial properties of nanomaterials such as carbon nanostructures which have been investigated thoroughly, the interactions of heteroatom-doped carbons with bacteria are to date poorly understood. Considering the fact that numerous purification and remediation wastewater treatments operate based on the activated carbon macroparticles, investigating the potential antibacterial properties of the heteroatom-doped bulk carbon materials as well as understanding the mechanisms of these reactions is of technological significance.

The aim of this study was to determine the impact of the Fe-, N- and/or S-co-doped porous carbon materials on the viability of various types of bacteria as well as on their affinity to these materials and to compare the results with corresponding metal-free heteroatom-doped carbons. The use of porous micrometric carbon particles (instead of nanostructures) allowed to minimize the effect of mechanical damage of cells, while exposing the bacteria to the studied carbon materials without addition of the external H_2_O_2_ eliminated the interference from ROS of extrinsic origin. In this paper, we demonstrate for the first time that the Fe–N–C type of heteroatom doped carbon macroparticles induces both the enhanced adhesion of bacteria to the carbon particle surface and the simultaneous loss of viability of the bacterial cells.

## 2. Materials and Methods

### 2.1. Fe–N–S Doped Porous Carbons

#### 2.1.1. Synthesis of the Studied Carbon Materials

The heteroatom-doped porous carbons were prepared according to the procedure presented by Kiciński et al. [22]. Briefly, resorcinol was condensed via the sol-gel method with selected aldehydes bearing either N-, S-, or O-containing heterocyclic rings. Since the sol-gel method was used, the obtained samples are referred to as gels. Polycondensation with 2-formylpyrrole yielded a N-doped carbon, condensation with 2-formylthiophene yielded a S-doped carbon, while condensation with 2-furaldehyde yielded an O-doped carbon (herein referred to as undoped carbon gel). An equimolar mixture of S- and N-containing aldehydes was used for dual S/N co-doping. Two batches of carbon samples were prepared: metal-free and Fe-doped carbon gels. To obtain metal-free samples the sol-gel polycondensation was induced by hydrochloric acid, while for the Fe-doping the anhydrous FeCl_3_ was used instead of the acid. In each synthesis, 2.0 g of resorcinol was used, and the molar ratio of resorcinol/aldehyde(s) was equal to 0.5. To prepare the metal-doped carbon samples, 4.0 g of FeCl_3_ was used. The obtained porous polymers (either with or without FeCl_3_) were pyrolyzed at 800 °C for 1 h. After the pyrolysis, samples obtained from mixtures with FeCl_3_ were leached in concentrated hydrochloric acid for 48 h to remove acid-accessible and leachable Fe-based fractions. Then, the samples were washed thoroughly with redistilled water and air dried. Acid leaching of the metallic phase is of crucial importance as it allows for production of stable Fe-doped carbons which do not release iron during further studies. As previously reported, the high temperature annealing of N- and Fe- co-doped organic matter yields materials where Fe is firmly bound to the char by formation of the carbon basal plane-embedded Fe–N_x_ complex structures such as single iron atom Fe(II)–N_4_–C structures [23], which are resistant to acid leaching. Finally, the metal-doped materials were subjected to subsequent annealing at 800 °C for 1 h. The obtained materials were labelled by symbols indicating the doped heteroatom(s); CN stands for solely N-doped carbon, CS for solely S-doped carbon, CNS for N/S-co-doped carbon, while C stands for carbon with no heteroatom doping. Samples denoted as CN-Fe, CS-Fe, CNS-Fe, and C-Fe are carbons additionally co-doped with iron.

#### 2.1.2. Characterization of the Carbonaceous Materials

The X-ray powder diffraction patterns (XRD) were registered on a Rigaku Ultima IV diffractometer using Co-Kα radiation (λ = 0.17890 nm) working in a parallel beam geometry, with 40 kV and 40 mA voltage and current, respectively. To determine the texture and the Brunauer-Emmett-Teller specific surface area (S_BET_) of the samples, the N_2_ adsorption measurements were performed at −195.8 °C using a Micromeritics ASAP 2020 apparatus. The total pore volume was estimated from the amount of N_2_ adsorbed at p/po ~0.99, while the volume of micropores was calculated by the *t*-plot method. To determine the wt.% content of S, N, C, and H elements an elemental analysis was performed using a Vario EL Cube apparatus (Elementar, Langenselbold, Germany). Please note that porous carbon always contains an extensive amount of oxygen and some of the doped carbon materials contained iron as well; consequently, the content of the determined heteroatoms does not add up to 100%.

### 2.2. Microorganisms and Media

The *Escherichia coli* (K-12 strain) and the *Bacillus cereus* (ATCC 11778 strain) were used. The bacterial medium in all experiments was Tryptic Soy Broth (TSB, Sigma-Aldrich, St. Louis, MI, USA) as a liquid medium and LB agar (Sigma-Aldrich, St. Louis, MI, USA) as a solid medium for the cultivation of bacteria.

### 2.3. Sorption Experiment

#### 2.3.1. Partition Coefficient

In order to assess the quantity of the carbon materials that yields significant differences in bacteria sorption, the partition coefficient was determined. At first, the suspensions of the bacterial cells were prepared using a sterile 0.9% NaCl. The initial density of the suspension was McFarland 4. The sorption experiments were conducted using sterile glass vials (15 mL). 5 mL of the bacterial suspension (10^9^ cfu mL^−1^) was added to the vials with a weighted portion of the doped carbon (5–30 mg). Next, the vials were tightly closed and horizontally mixed (50 rpm) for 3 h at 25 °C. After the sorption, 1 mL of the solution was centrifuged for 2 min at 2600× *g* with 0.3 mL of glycerol (Avantor Performance Materials, Poland S.A., Gliwice, Poland) to separate the carbon material with adsorbed cells from unadsorbed bacteria. A similar procedure was previously implemented in the study on bacterial sorption on nanomaterials [24]. The absorbance of both 150 µL of supernatant and the bacterial suspension before the sorption were measured at 405 nm using a microplate reader (Thermo, Multiscan FC, Waltham, MA, USA). All experiments were conducted in triplicate, and the measurements were carried out simultaneously using 96-well sterile microplates. The blanks were both a sterile 0.9% NaCl solution and a supernatant after the sorption experiment without the bacterial suspension (the doped carbons in the salt solution only). The dependency between the absorbance of the bacterial suspension used and the quantity of bacterial cells was previously determined using the agar plate method after serial dilution of the bacterial suspension in a sterile 0.9% NaCl solution.

The partition coefficient of the bacteria after the sorption was calculated using the formula:

P = (C_s_/C_eq_), where:
C_s_—is the concentration of the bacteria in solid phase after sorption, and C_s_ = C_o_ − C_eq_C_o_ —is the initial concentration of bacteriaC_eq_—is the concentration of the bacteria in liquid phase after sorption.


#### 2.3.2. Sorption Isotherms

The isotherms of bacteria adsorption on doped carbons were obtained conducting the same experiments as described in the Section 2.3.1, yet using a particular mass of the carbon sorbent (20 mg) and four different concentrations of bacteria (0–10^9^ cfu mL^−1^). The relationship between number of adsorbed cells (*n* = (C_o_ − C_eq_)V/m [cells g^−1^]) and the equilibrium concentration of the bacteria remaining in the suspension after the sorption (C_eq_ [cells mL^−1^]) was plotted and interpolated using the Langmuir isotherm *n* = (A_m_·K·C_eq_)/(1 + K·C_eq_), where:
A_m_ is the maximal number of adsorbed bacteria;K is the Langmuir constant;C_eq_ is the equilibrium bacterial concentration.


### 2.4. Loss of Viability

The 30 mg portion of each doped carbon material was mixed with 5 mL of the bacterial suspension (McFarland 4) in a sterile glass vial (15 mL) and gently, horizontally shaken (50 rpm) for 1 h at 25 °C. Next, 1 mL of the suspension was transferred to an Eppendorf tube (2 mL) and inoculated with 100 µL of the propidium iodide solution (2 mg 0.1 L^−1^, pH = 7.4). After 5 min., 50 µL of the acridine orange solution (2 mg 0.1 L^−1^, pH = 7.4) was added. The stained suspension (20 µL) was placed on the microscopic slide with a cover and analyzed using the epifluorescence microscope fitted with the blue filter (DM500 filter with a band-pass 460–490 nm excitation filter). The captured images yielded living bacteria as green, while dead or damaged organisms as red. The loss of viability of the free living bacteria and bacterial cells adsorbed on the surface of the carbon particles was computed using data collected from images with the following formula: Loss of viability = (red stained cells/total cells) × 100%. The control viability was measured using the bacteria incubated without doped carbons.

### 2.5. Activity of Dehydrogenases

The metabolic activity of the bacteria was determined based on the dehydrogenases activity using the method previously successfully tested and reported by Borkowski et al. [25]. In the experiment, the 12-h cultures of *E. coli* and *B. cereus* bacteria with the growth densities of 0.594 and 1.24 equal to 227 µg of protein mL^−1^ and 230 µg of protein mL^−1^, respectively, were used. The protein amount was determined by the Lowry method with a previously tested modification [24]. In the first step of the procedure, 2.5 mg portion of each doped carbon material and 0.5 mL of the bacteria culture were placed in a sterile Eppendorf tube and gently mixed. The suspensions were incubated at 25 °C for 2 h, then inoculated with 0.1 mL of 3% triphenyltetrazolium chloride (TTC, Avantor Performance Materials Poland SA, Gliwice, Poland) and allowed to react for another 15 min. at the same temperature. After this, the suspensions were centrifuged at 12,000× *g* for 30 s, and the supernatants were discarded. The red formazan (TPF) was extracted from pellets using 0.5 mL of 96% ethanol (Avantor Performance Materials Poland SA, Gliwice, Poland). Next, the extracts were centrifuged again, and 0.15 mL of supernatant were placed on the 96-well microplate for the absorbance measurements (at 405 nm). The results were presented as a change of absorbance during a min. in relation to mg of protein (dA_405_ min^−1^ mg of protein^−1^). The control measurements were conducted on bacteria cultures without carbon materials. The possible effect of the heteroatom-doped carbon particles on the absorbance measurements was assessed using the carbon suspensions in a sterile TSB medium.

### 2.6. ICP-MS Analysis

In order to determine the potential effect of the soluble metallic impurities on the bacteria viability, the chemical composition of the heteroatom-doped carbon suspensions was analyzed. The doped carbon was placed in a falcon tube (15 mL) and suspended in deionized water (30 mg 5 mL^−1^). Next, after vigorous mixing, the tubes were left for 3 h and centrifuged at 3000× *g* for 5 min. The supernatant was analyzed for metal content using the ICP-MS technique (with an ELAN 6100 Perkin Elmer instrument).

### 2.7. Statistical Analysis

For the data analysis, the Statistica 10 software (StatSoft Inc., Tulsa, OK, USA) was used. In order to fit the experimental data to the Langmuir isotherms the method of least number of squares for nonlinear models was used. The other data were analyzed for the significance of mean differences using one-way analyses of variance (ANOVA) at *p* < 0.05. Post-hoc tests for pair-wise differences and the identification of homogeneous subgroups were conducted using Tukey’s HSD procedure.

## 3. Results

### 3.1. Physicochemical Characterization of the Doped Carbons

The obtained metal-free and metal-containing carbon materials differed significantly depending on the heteroatom dopant. Table 1 summarizes the content of heteroatoms and textural properties of the studied samples, indicating some general trends. The Fe-doping yielded materials with the low surface area, the low heteroatom content, and the high carbon content, unless sulphur was present as a co-dopant. Whereas Fe-doped yet S-free samples (C-Fe and CN-Fe) were highly graphitized carbons with reduced porosity, the presence of sulphur blocked the graphitization during pyrolysis yielding materials of the greater surface area, the greater content of heteroatoms and with disordered carbon structures (samples CS-Fe and CNS-Fe), see XRD patterns in Figure 1 and Table 1 for details. The Fe-free carbons were all of disordered structure (they did not undergo graphitization during pyrolysis) and they contained a substantial amount of the doped non-metallic heteroatoms. The samples containing sulphur exhibited high S_BET_ values of up to 800 m^2^g^−1^, while the lowest S_BET_ values c.a. 200 m^2^g^−1^ were determined for the Fe-doped but S-free samples. The N_2_ adsorption-desorption isotherms of the studied carbon samples (presented in Appendix A, Appendix A) suggested that the S-doped samples were exclusively microporous materials (CS-Fe exhibited only a slight presence of mesopores), while the CN-Fe and C-Fe carbon were highly mesoporous materials. As commonly known, iron dissolves elemental carbon during the high-temperature annealing and acts as a catalyst of carbon crystallization (graphitization). The amorphous carbon materials are usually highly porous and contain heteroatom impurities (O,N,H, etc.). Graphitization of carbon caused by its high-temperature dissolution in the presence of iron followed by its precipitation from the Fe–C mix yielded substantial reduction of microporosity and expulsion of heteroatoms from the produced graphitic phase. As a result, the Fe-containing samples presented in Table 1 exhibited low content of heteroatoms and low S_BET_ values. However, in case of the carbon sample also containing sulphur, iron reacted with sulphur yielding FeS, which did not dissolve carbon. For this reason, the S-dope carbon materials did not undergo Fe-catalyzed graphitization and hence microporous carbon was obtained upon acid leaching of the FeS phase. These phenomena were discussed in detail in some earlier reports [22]. The SEM analysis revealed that the studied carbon materials were of a very similar morphology. All samples consisted of colloidal microparticles about ~2 µm in size (the image of the Fe- and N-co-doped carbon is shown as a representative example). This morphology of interconnected spherical microparticles is typical for sol-gel derived porous carbon [26]. The carbon samples studied herein were materials with micrometric features (they did not consist of nano-objects) and their “nano” character concerned only their inner nanoporosity.

### 3.2. Adsorption of Bacteria

The results indicated that the studied carbons differed significantly regarding the sorption of bacterial cells. Furthermore, sorption depended on the type of bacteria (Gram-positive and Gram-negative bacteria). Firstly, the partition coefficient (*P*) was analyzed in relation to the adsorbent weight. *P* was expressed as a ratio between the bacteria remaining in suspension after sorption and the bacteria adsorbed on carbon material. Considering the slope of the linear relationship between the mass of the adsorbent and the partition coefficient (Figure 2), the strongest dependency was observed for *E. coli* and CN-Fe and CN carbons. Starting from the highest value of the slope parameter of the above-mentioned linear relationship, the studied materials can be ranked as follows: CN-Fe > CN > CS-Fe > CNS-Fe > CNS > C > C-Fe > CS, which is relevant to the following values of the derivative of a linear function: 580, 296, 23, 15, 3.4, 2.9, 2.2 and −3.4 P g^−1^. This indicated that the CNS, C, C-Fe and CS samples basically did not change the partition coefficient despite the increase in the mass of the adsorbent used. In the case of *B. cereus*, the highest slope parameters were observed for CN-Fe and C-Fe, which is the main difference in relation to the previous *E. coli* system, where the C-Fe material adsorbed bacteria virtually independently of the adsorbent mass. The rest of the doped carbon materials also exhibited different properties in presence of *B. cereus* in comparison to *E. coli* adsorption results. Considering the slope parameters of the mentioned linear regression, in the case of *B. cereus*, the studied carbon materials can be ranked as follows: CN-Fe > C-Fe > CN > CNS-Fe > C > CS-Fe > CS > CNS, which is relevant to the following values of the derivative of a linear function: 125, 113, 62, 37, 13, 11, 9.1, and 4.7 P g^−1^.

The *p* analysis indicated that, in the studied system, an adsorbent mass above 20 mg resulted in significant differences between bacteria sorption properties, hence this amount was used in the following isotherm sorption experiments. The obtained isotherms (Appendix A, Appendix A) indicated the Langmuir equation as the model of the cell adsorption on the doped carbon materials, yielding correlation coefficients in the 0.96–0.99 range. The obtained Langmuir isotherms allowed to assess the maximum sorption capacity based on the obtained A_m_ (Table 2). The most efficient carbon materials in terms of bacteria adsorption were the CN and CN-Fe doped carbon samples. The maximum sorption of *E. coli* was 3.9 and 4.7 × 10^10^ cells g^−1^ for CN and CN-Fe, respectively. The maximum sorption of *B. cereus* was 65 and 195 × 10^10^ cells g^−1^ for CN and CN-Fe, respectively. Furthermore, in the case of *E. coli*, the Fe-doped materials were able to adsorb the cells more efficiently than other samples (C, CS, CNS). It is worth mentioning that the CN-Fe sample exhibited the greatest adsorption capacity towards both bacteria types.

### 3.3. Loss of Viability

The adsorption of bacteria on the doped carbon surface may potentially lead to the cell damage, both mechanically and due to the reactive oxygen species activity. Accordingly, the viability test was conducted to evaluate whether damage indeed occurs (Figure 3). Interestingly, neither the dopant-free carbon (the C sample) nor the N-, S-, and N/S-co-doped metal-free materials caused the visible effect on the bacteria viability; the cells exposed to the life-death staining were green under fluorescence microscope. The obtained results concerned both free living and adsorbed cells. Only a slightly enhanced loss of viability was observed for the adsorbed bacteria, however, the differences were not significant unlike the results obtained for the Fe-doped carbons. Generally, the presence of Fe increased the loss of viability of the adsorbed bacteria in comparison to the control. No significant differences were found only in the experiment with the CS-Fe carbon. However, the effect was different depending on the type of the bacteria and the presence of the coexisting dopants. Furthermore, Fe-doped carbons increased the loss of viability not only of adsorbed but also of free-living *B. cereus*. This effect was not observed for *E. coli*, for which the loss of viability in the control and among the free living cells was ca. 5%, while the loss of viability of the cells on the surface of CN-Fe carbon reached 25%. In the case of *B. cereus*, much greater damage was found (ca. 40% loss of viability), and in these terms the most toxic materials were CN-Fe and CNS-Fe. The Fe- and N-co-doped carbons caused the greatest loss of viability.

The analysis of the chemical composition of the suspensions used in the experiments revealed extremely low concentrations of metals (Appendix A). The maximum detected concentration of the potentially toxic metals concerned Cu and Cr; however, the obtained values did not exceed 0.1 mg L^−1^. Furthermore, in terms of the chemical composition, the suspensions were statistically identical within the analytical error of 5%. The results indicated that the Fe-doped carbons were stable and did not release iron ions under the experimental conditions. The iron appeared to be permanently bound to the structure of the carbon materials.

### 3.4. Dehydrogenase Activity

The incubation of *E. coli* bacteria with doped carbon materials revealed a decrease in dehydrogenase activity in the case of Fe-doped samples, and the differences among type of dopants used were statistically significant (Figure 4). For the control culture, 0.085 dA min^−1^ mg of protein^−1^ was determined, while for the cultures with C-Fe and CS-Fe the results did not reach 0.01 dA min^−1^ mg of protein^−1^. In the cultures with CN-Fe and CNS-Fe the measured activity was also lower in comparison to the control (ca 0.06 dA min^−1^ mg of protein^−1^ in both cases). Generally, the *B cereus* cultures yielded lower dehydrogenase activity, yet some variability among the samples was observed. The S-doped and Fe-doped materials decreased the activity of dehydrogenases significantly; the values did not reach 0.005 dA min^−1^ mg of protein^−1^ while for the control 0.03 dA min^−1^ mg of protein^−1^ was determined. However, it seems that the N-dopant caused either no significant increase in activity or compensation of the effects caused by the Fe- and S-dopants. The most significant increase in activity (0.06 dA min^−1^ mg of protein^−1^) was determined for the CNS-Fe carbon.

## 4. Discussion

It has been already reported that the N-doped carbon materials can adsorb bacteria more efficiently in comparison to the non-doped carbon [26], which indicated possible applications of these materials in the water treatment technology. Carbon materials are widely used as sorbents in disinfection systems on both home and technological scale [27] and some recently published studies showed, that dopants can significantly alter the technological properties of the carbon materials [28,29]. However, as it was also highlighted [26], the adhesion of bacteria to the N-doped carbon leads to the biofilm formation which is detrimental in the water treatment systems. Our present studies confirm that the N-doped carbon indeed adsorbs bacteria efficiently and while this effect is minimized by the S-codopant, the Fe-dopant slightly increased the bacteria sorption. This is most likely due to the change in the wettability of the doped carbon surface, as previously suggested [26]. However, in terms of the technology and material properties, the important question is whether the adsorption of the bacteria is correlated with their loss of viability (bactericidal effect), as such materials are of a high import in many branches of the bio- and wastewater treatment technologies. Our results indicated that neither N- nor N/S-co-doped metal-free carbons caused the visible loss of the bacteria viability, yet the presence of a Fe-dopant significantly increased the cell damage. Furthermore, for the Fe-doped carbon, the enhanced adhesion of both Gram-negative and Gram-positive bacteria was observed. Thus, a kind of synergistic effect lowering the bacteria viability occurred on the Fe- and N-co-doped materials, and the process was highly dependent on the mass of the carbon sorbent. It was especially visible for the experiments with the Gram-positive *B. cereus*, where the greatest bacteria sorption was determined for the CN-Fe sample (see Appendix A). Furthermore, the loss of viability was significantly greater for the Fe-doped carbon materials than for their metal-free counterparts. However, it should be emphasized that the S-dopant probably bears the antagonistic effect. Therefore, on the one hand the Fe-N-co-doped carbon showed unequivocally both the enhanced bacteria sorption and the enhanced loss of viability yet, on the other hand, the metal-free S- or N-doped carbon materials presented only efficient bacteria sorption. Such versatile carbon-based materials can find a variety of applications e.g., in water treatment where the sorption and bactericidal effect are required, or e.g., in bioreactors where the carbon beds should promote biofilm formation without damaging the bacterial cells. The studied materials can potentially be considered as the new functional materials able to adsorb and damage bacterial cells simultaneously, regardless of the lack of nano-sized features or of the additional extrinsic chemical agents introduced to the solution.

The presented results clearly show that the greatest loss of viability occurred for the bacteria adsorbed on the surface of the Fe-containing carbon. Hence, we hypothesize that the presence of Fe in the carbon materials is essential for these phenomena. In this regard, an intriguing question arises: why did the Gram-positive bacteria turn out to be more sensitive than the Gram-negative ones? If the mechanical damage was the key factor for the deteriorating bacteria–carbon interactions, then the Gram-positive bacteria should be more resistant to this process. On the other hand, the Gram-negative bacteria are usually more resistant to the polar chemical factors (e.g., ions, ionic liquids, etc.) due to their outer lipid membrane. Nevertheless, as demonstrated elsewhere, despite structural differences in the Gram-positive and Gram-negative bacteria cell walls, their resistance against mechanical damage is not very straightforward and depends strongly not only on the wall structure but also on the specific experimental conditions [30]. Furthermore, as presented in this study: (i) the loss of viability of the free living Gram-positive bacteria was enhanced in the presence of Fe- and N-co-doped materials; (ii) the ICP-MS analyses revealed that the carbon suspensions did not contain toxic metals of bactericidal concentrations; and (iii) both types of bacteria reacted to the presence of Fe-doped carbon in a similar way, showing a significant decrease in dehydrogenase activity. Consequently, the mechanical damage is not a key factor of the process, and most likely some chemical parameters (other than chemical impurities from suspensions) play a pivotal role in the studied system. Extensive literature data indicate that the Fe–N molecular centers (e.g., iron–nitrogen coordination moieties) built into the carbon structure are able to activate O_2_ molecules in a similar manner to metalloenzymes, yielding ROS such as H_2_O_2_ or superoxide radicals, which in turn can initialize subsequent reactions towards more reactive hydroxyl radicals. However, the precise mechanism of this catalytic activation of atmospheric oxygen is still under debate. Chen et al. [3,19] suggested that Fe–N–C carbonaceous materials (N-doped carbons with Fe nanoparticles and/or Fe–N_x_ coordination sites) can catalyze O_2_ reduction to ^•^O_2_^−^ and to singlet oxygen (^1^O_2_). They showed that O_2_ can be reduced by organic compounds to yield H_2_O_2_. The hydrogen peroxide reacting with the Fe^2+^/Fe^3+^ in the Fe–N_x_–C structure can generate superoxide radicals. H_2_O_2_ can also react with the Fe–N_x_ centers and produce ^•^OH by a Fenton-like reaction. However, regardless of the possible reaction pathways yielding ROS, the key role of the initial bonding of oxygen by the Fe–N_x_ centers in the carbon matrix seems to be confirmed experimentally [3] and our results are in line with these findings. The Fe- and N-co-doped carbon materials [3] were shown to catalyze oxidation of tetramethylbenzidine (TMB) and their catalytic activity was inhibited under the N_2_-saturated condition yet enhanced under the O_2_-saturated condition. Moreover, the TMB was oxidized without H_2_O_2_ addition, which clearly suggests that the Fe–N_x_ center can directly activate O_2_ exhibiting oxidase-like activity. Our results also indicate that loss of the bacteria viability occurs in the presence of Fe- and N-co-doped carbons without any H_2_O_2_ addition. In previous reports, the bactericidal effect was shown using Fe- and N-co-doped carbon nanomaterials but hydrogen peroxide was also added at minimal, physiological concentration suggesting the important role of H_2_O_2_ [31]. It seems that the Fe- and N-co-doped carbon materials can significantly enhance the bactericidal effect of hydrogen peroxide, even if its concentration is much lower than that used for clinical purposes [31]. If the bacteria are exposed solely to the external H_2_O_2_, or solely to the Fe/N-co-doped carbon, survival rates are high [4]. A comparison of the oxidase- and peroxidase-like activities of Fe–N–C materials showed that the oxidase-like activity is inferior to peroxidase-like activity [5]. Furthermore, the Fe–N–C materials can also catalyze decomposition of H_2_O_2_ generated by the studied cells yielding ^•^OH. Our results revealed that the addition of hydrogen peroxide is unnecessary to cause a significant toxic effect, even with the bulk (non-nanometric) Fe- and N-co-doped carbon particles. The Fe–N_4_ center is considered as the most active site which activates O_2_ at room temperature to generate various ROS (e.g., ^1^O_2_, O_2_^•−^, and H_2_O_2_). Some researchers assume that Fe–N–C materials activate O_2_ only to ^1^O_2_, O_2_^•−^, and H_2_O_2_, instead of more aggressive ^•^OH [32]. While most of the known artificial enzymes are not able to use O_2_ directly from the solution and the H_2_O_2_ must be added as an oxygen donor, the Fe–N–C structures are highly heterogeneous materials with a variety of different active centers—some of which show peroxidase—while others show oxidase-like activity able to reduce either H_2_O_2_ and/or O_2_ simultaneously [33]. Indeed, it is important to highlight that Fe–N–C nanozymes usually show multienzyme activities [34,35]. However, incorporation of the Fe–N_4_ sites into graphene allowed to obtain the material with an exclusive peroxidase-like activity [34]. As demonstrated, the Fe–N_4_ sites incorporated into graphene sheets exhibited low reaction energy for H_2_O_2_ but the O_2_ dissociation reaction over the Fe–N_4_ was endothermic, which indicated selective peroxidase-like activity. To solve the problem of nanozymes poor selectivity, the metal-organic framework/glucose oxidase hybrids were constructed [36]. Such materials allow to oxidize glucose via glucose oxidase and O_2_ and the produced hydrogen peroxide is subsequently decomposed by the MOF possessing peroxidase activity to generate toxic ^•^OH. This bifunctional MOF/glucose oxidase hybrid was tested as the antibacterial agent against formation of biofilms [36]. Generally, it can be assumed that Fe–N–C materials facilitate reduction of O_2_ by catalyzing the transfer of electrons and protons from organic matter to oxygen. In the first step O_2_ can be reduced to O_2_^•−^ and to H_2_O_2_. Then, other ROS such as ^•^OH can be released by the so-called heterogeneous Fenton process [37]. The artificial enzymes with oxidase-/peroxidase-like activities are getting increased attention as the materials preventing biofouling. For instance, the Fe_3_O_4_ nanoparticles with peroxidase-like activity enhanced oxidative cleavage of biofilm components in the presence of H_2_O_2_, which prevented biofilms formation and exterminated both planktonic bacteria and microbes residing within the biofilm [38]. In other interesting research, the combination of iron oxide or iron sulphide nanozymes and bacteria generating biogenic H_2_O_2_ was used to eliminate oral biofilms [39]. Our results showed that the obtained Fe- and N-co-doped porous carbon materials possess similar, intriguing properties.

## 5. Conclusions

N-doping significantly enhances the affinity of Gram-positive and Gram-negative bacteria to both metal-free and metal-containing porous carbons. From the technological point of view, the N-coordinated iron-doped carbons (Fe–N–C carbonaceous materials) exhibit important properties for at least two reasons: (i) the N- and Fe-co-doped carbons strongly enhance the adhesion of bacterial cells, so these materials can potentially be implemented in various technologies such as water treatment; (ii) the Fe-doped carbons reveal significant bactericidal effects, which in combination with the enhanced bacterial cell sorption may allow for production of new, advanced multifunctional materials. Such carbon materials could perform a double function, that is to adsorb bacteria and significantly reduce the biofilm formation. Today, most microbiological filters operating based on the activated carbon undergo quick biofouling which leads to their rapid deactivation. Utilization of Fe- and N-co-doped porous carbons could solve this problem. Additionally, it should be highlighted that the studied Fe-N-co-doped carbon materials are made up of large micrometric colloids (unlike nanostructures/nanoobjects) and hence are probably not dangerous to humans. The Fe- and N-co-doped carbons are highly heterogeneous materials with various active centers [33]. It is hypothesized that some of the active centers show peroxidase while others show oxidase-like activity to reduce either H_2_O_2_ or O_2_, and in this term, the heterogeneity of the presented materials contributes to the complexity of their bactericidal effects. This study showed striking similarities but also striking differences between the N-doped, metal-free and the N- and iron co-doped carbons. Both materials exhibit high affinity for bacteria sorption yet Fe co-doping yields bactericidal properties without addition of any external sources of ROS-generating chemicals. By using the carbon micron-sized particles we eliminated the possibility of bactericidal effects caused by the mechanical cell wall disruption (typical for carbon nanostructures). The influence of transition metal impurities potentially released from the carbon samples was also excluded. Consequently, the loss of bacteria viability via their contact with the Fe–N–C materials is solely due to the intrinsic, specific properties of carbon caused by its simultaneous doping with nitrogen and iron. While the N-doping enhances the affinity of bacteria to carbon surface, the Fe-doping induces additional bactericidal effects.

## Figures and Tables

**Figure 1 materials-13-03707-f001:**
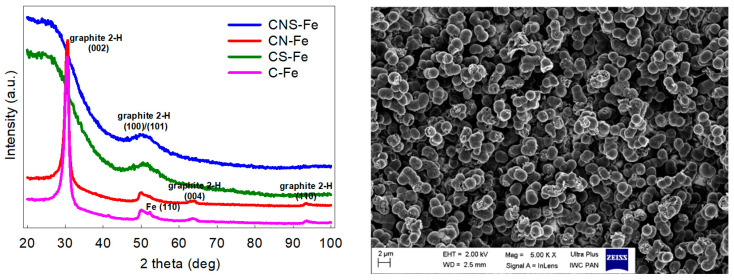
The XRD patterns of the Fe-containing carbon materials. Samples without sulphur underwent significant graphitization (CN-Fe, C-Fe) while sulphur-doped carbons (CS-Fe, CNS-Fe) remained disordered. The representative SEM microphotograph of the studied materials (the Fe- and N-co-doped carbon sample).

**Figure 2 materials-13-03707-f002:**
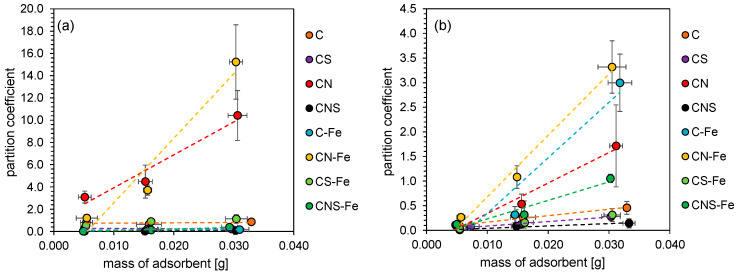
The partition coefficient of *E. coli* (**a**) and *B. cereus* (**b**) after sorption at different masses of doped carbon.

**Figure 3 materials-13-03707-f003:**
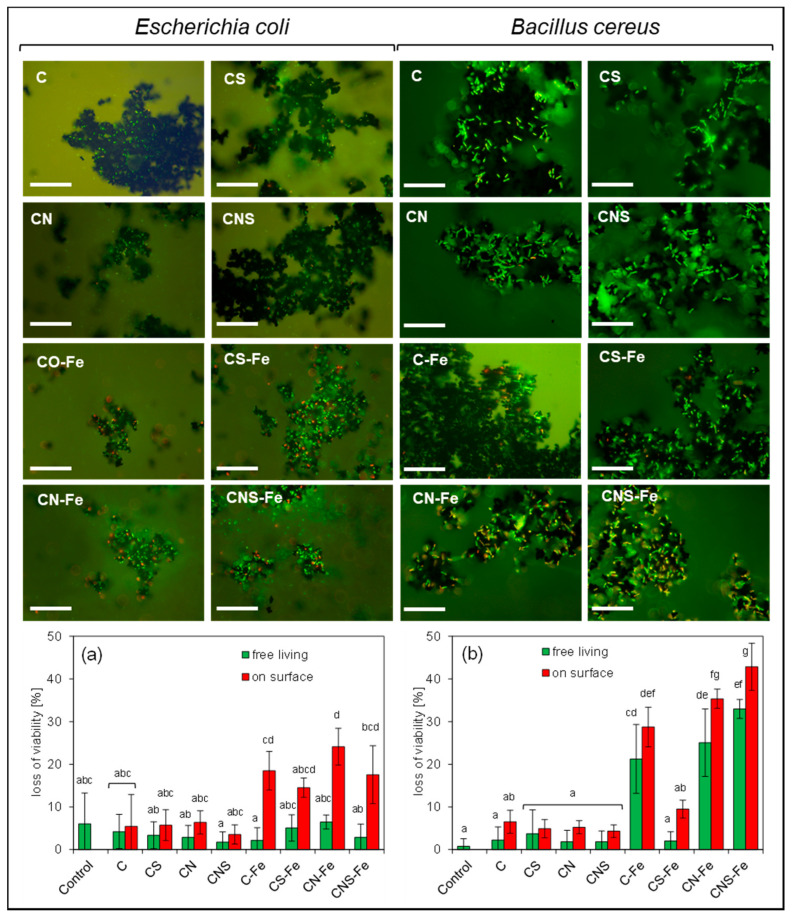
Examples of fluorescent microscope pictures, bar 50 µm. Loss of viability of free living and adsorbed bacteria on the doped carbons, (**a**) *E. coli* and (**b**) *B. cereus*. The bars with the same letters do not differ statistically.

**Figure 4 materials-13-03707-f004:**
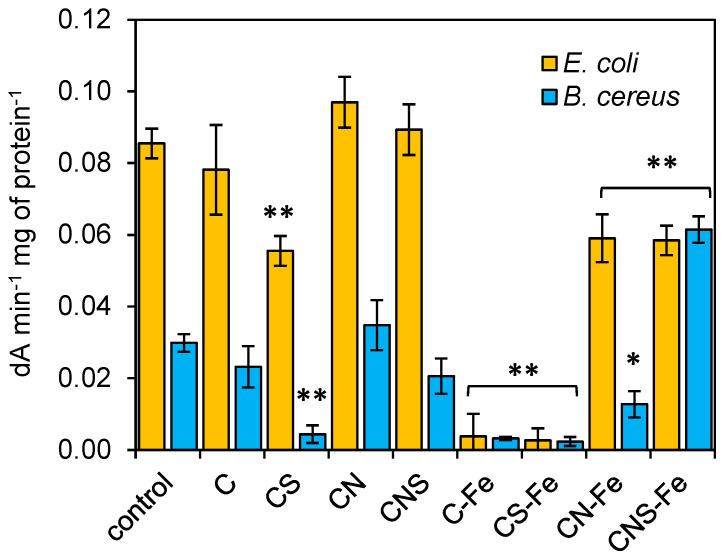
Activity of dehydrogenases after incubation of bacteria with the doped carbons. Significant difference (* at *p* < 0.1 and ** at *p* < 0.05) compared to the control and standard deviation are marked.

**Table 1 materials-13-03707-t001:** Elemental composition (wt.%) and textural parameters of the studied carbon samples.

Sample	N	C	H	S	S_BET_ (cm^3^ g^−1^)	V*_t_* (cm^3^ g^−1^)	V_mic_ (cm^3^ g^−1^)
C-Fe	0.00 ± 0.00	98.11 ± 0.81	0.13 ± 0.03	0.12 ± 0.07	165	0.40	0.02
CS-Fe	0.21 ± 0.04	81.79 ± 0.74	0.41 ± 0.05	9.88 ± 1.08	800	0.41	0.34
CN-Fe	1.02 ± 0.06	96.01 ± 0.32	0.27 ± 0.02	0.00 ± 0.00	225	0.51	0.06
CNS-Fe	5.04 ± 0.09	82.33 ± 0.95	0.99 ± 0.12	1.54 ± 0.38	535	0.25	0.23
C	0.00 ± 0.00	93.92 ± 0.21	0.41 ± 0.07	0.00 ± 0.00	670	0.26	0.24
CS	0.00 ± 0.00	82.24 ± 1.21	0.21 ± 0.02	11.01 ± 1.77	700	0.27	0.25
CN	6.66 ± 0.48	80.18 ± 1.31	1.34 ± 0.37	0.00 ± 0.00	290	0.13	0.11
CNS	4.11 ± 0.31	82.83 ± 0.81	0.47 ± 0.06	5.31 ± 0.56	530	0.20	0.19

S_BET_ is the specific surface area, V*_t_* is the total pore volume, V_mic_ is the volume of micropores.

**Table 2 materials-13-03707-t002:** The Langmuir isotherm parameters based on the adsorption of the bacteria on the doped carbons.

		*E. coli*			*B. cereus*	
Sample	A_m_ (×10^10^ Cells g^−1^)	K	r	A_m_ (×10^10^ Cells g^−1^)	K	r
C	0.72	5.21	0.99	29.0	0.12	0.99
CS	0.70	5.60	0.99	7.16	0.26	0.99
CN	3.88	18.1	0.99	64.6	0.94	0.99
CNS	0.95	3.32	0.99	4.89	0.11	0.99
C-Fe	1.49	4.33	0.99	14.7	0.10	0.97
CS-Fe	1.47	35.1	0.98	37.2	0.47	0.98
CN-Fe	4.74	41.9	0.96	195	0.59	0.98
CNS-Fe	1.47	2.69	0.99	47.1	0.05	0.99

A_m_ is the maximal number of adsorbed bacteria (× 10^10^ cells g^−1^), K is the Langmuir constant, *r*—correlation coefficient.

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
