# Peer review of "Interactions of Fe–N–S Co-Doped Porous Carbons with Bacteria: Sorption Effect and Enzyme-Like Properties"

_materials, 2020, doi:10.3390/ma13173707_

Round 1

Reviewer 1 Report

Borkowski and colleagues performed prepared a series of metal-free and metal-containing N -S doped porous carbons. They performed a through compositional and textural characterization on these carbons. Finally, they evaluate performance of carbons and role of metal and non-metal dopants on  the bacteria adsorption.

This is a very systematic and organized manuscript. I would recommend publishing after applying minor adjustment and answering these questions below.

Line 26: The sentence starting with “various” is not clear and needs to be rephrased.

Table 2: The expectation is that the BET surface area for undoped carbon to be the highest. Authors should explain why the BET surface area for CS-Fe with two bulky dopants is higher than bare C.

The Nitrogen isotherm should be provided in the main manuscript.

I would highly recommend that authors mention the typical size of bacteria studied in this research. Then if they can correlate the bacteria size with the average pore size a conclusion can be drawn just based on the porous structure of carbon. If there is any similar study in this regard, authors can cite it.  

Author Response

Responses to the Reviewers’ comments (manuscript ID materials-896124)

We would like to thank the Editor and all Reviewers for the effort and time they invested to process and improve our manuscript. We carefully corrected the manuscript according to all of the given suggestions and recommendation. For convenience, the most important changes are marked up in the revised article.

Reviewer 1

Line 26: The sentence starting with “various” is not clear and needs to be rephrased.

RESPONSE: We rephrased this sentence to make it easier to understand (we marked the changes in the manuscript), and added appropriate reference supporting this particular claim (Ding, C., Yan, Y., Xiang, D. et al. Magnetic Fe3S4 nanoparticles with peroxidase-like activity, and their use in a photometric enzymatic glucose assay. Microchim Acta 183, 625–631 (2016). https://doi.org/10.1007/s00604-015-1690-6)

Table 2: The expectation is that the BET surface area for undoped carbon to be the highest. Authors should explain why the BET surface area for CS-Fe with two bulky dopants is higher than bare C.

RESPONSE: Yes, this is an interesting phenomenon; we added appropriate elucidation in the revised manuscript. Iron can dissolve carbon and hence it acts as a good graphitization catalyst. For this reason, polymers and chars doped with iron undergo extensive graphitization during high-temperature annealing (conversion of amorphous and often microporous carbon into crystalline and much less porous graphitic structure). Graphitization of carbon via dissolution of amorphous and precipitation of crystalline carbon over Fe phase also causes removal of heteroatoms from carbonaceous phase (e.g. O, H, N) producing pure graphite (graphitic phase). However, if sulfur is present in carbonaceous material it reacts instantaneously with iron to give iron sulfide. Carbon does not dissolve in iron sulfide and hence cannot undergo catalytic graphitization. For this reason, Fe- and S-co-doped amorphous carbon is not converted into graphite and heteroatoms stay in the carbon scaffold. Consequently, after acid leaching of iron mineral phases (sulfides) one obtains amorphous, microporous carbon rich in heteroatoms. We have explained this phenomenon in detail in our previous work, which we cite in the current manuscript for reference (Kiciński, W.; Dembinska, B.; Norek, M.; Budner, B.; Polański, M.; Kulesza, P.J.; Dyjak, S. Heterogeneous iron-containing carbon gels as catalysts for oxygen electroreduction: Multifunctional role of sulfur in the formation of efficient systems. Carbon 2017, 116, 655–669, doi:10.1016/j.carbon.2017.02.045.).   

The Nitrogen isotherm should be provided in the main manuscript.

RESPONSE: We agree that the shape of N2 adsorption-desorption isotherms can give valuable insight into the texture of porous materials. However, in order to keep the manuscript concise, we eventually decided to add the isotherms in the Supplementary Materials.

I would highly recommend that authors mention the typical size of bacteria studied in this research. Then if they can correlate the bacteria size with the average pore size a conclusion can be drawn just based on the porous structure of carbon. If there is any similar study in this regard, authors can cite it.  

RESPONSE: This is an interesting issue to consider. The typical sizes of the studied bacteria are about ~2 µm for E. coli and up to 3 - 5 µm for B. cereus, respectively. Since a bacterium is a quite “large” object from the point of view of nanopores occurring in activated carbons, carbon gels, mesoporous carbons, etc. there is no correlation between carbon porosity measured by cryogenic N2 physisorption and bacteria adsorption capacity. This is because N2 physisorption measures porosity of pores only up to ca. 100 nm and the SBET originates only prom this porosity. It is rather the physicochemical nature of carbon surface (wettability, functional groups, acidity, etc.) that determine bacterium‒carbon affinity. But, importantly nono-porosity and SBET values do affect physical properties such as wettability, functional groups, acidity, etc. So there is indirect correlation but it is quite difficult to quantify and quality directly.

Reviewer 2 Report

In this manuscript, the authors investigated antibacterial effects using metal-free N and/or S doped or metal-containing Fe−N−S 17 doped porous carbons. Overall, the manuscript is interesting. However, minor revision is needed before Materials can consider it.

Introduction

Line no 57 -59: The sentence starts with; hence it is possible that….  The texts are not clearly stated, and it should be revised.

Line no 60 -63: the sentences should be rephrased and properly presented; one or two references correlate to work could be added.

Please clearly indicate the aim of the study at the end of the introduction.

Discussion: Overall few more reference correlate to work could be added.

Conclusion

Please conclude the significant findings of this study.

Languages: Grammatical error and some mistakes are founded. Please carefully rechecked and revised.

All the reference should be checked thoroughly and corrected.

Author Response

Responses to the Reviewers’ comments (manuscript ID materials-896124)

We would like to thank the Editor and all Reviewers for the effort and time they invested to process and improve our manuscript. We carefully corrected the manuscript according to all of the given suggestions and recommendation. For convenience, the most important changes are marked up in the revised article.

Reviewer 2

Line no 57 -59: The sentence starts with; hence it is possible that….  The texts are not clearly stated, and it should be revised.

RESPONSE: Indeed, we agree. We revised this phrase to make it clearer (we marked the changes in the manuscript).

Line no 60 -63: the sentences should be rephrased and properly presented; one or two references correlate to work could be added.

RESPONSE: We agree, also this part was not clear enough. We revised this part to make it clearer (we marked the changes in the manuscript).

 Please clearly indicate the aim of the study at the end of the introduction.

RESPONSE: As suggested, we stated the aim of the study at the end of the introduction.

Discussion: Overall few more reference correlate to work could be added.

RESPONSE: As suggested, we added some new, interesting and in our opinion highly-relevant references, which correlate to our work (6 new references were added).

Please conclude the significant findings of this study.

RESPONSE: At the very end of the conclusion we added elucidation of the significance of this work.

Languages: Grammatical error and some mistakes are founded. Please carefully rechecked and revised. All the reference should be checked thoroughly and corrected.

RESPONSE: We carefully revised the language of our manuscript; indeed, there were some errors in the text and also in references. Thank you for very careful and thorough review.

Reviewer 3 Report

This work is well-organized and the results have the potential to be interesting and useful. I recommend publishing “after minor revision” this manuscript in the Materials journal. However, from my point of view, there are a few questions and suggestion for correction that should be addressed before its publication.

Why is there such a big difference between the CNS compositions of ferrous and non-ferrous samples? For example in the case of nitrogen, the N-content in sample CN-Fe is 1.02%, while in the case of CN it is 6.66%, which is more than six times bigger. Or in the case of the Sulphur content: CNS-Fe 1.54% but CNS 5.31%, which is five times bigger. This phenomenon must be explained in the 3.1 session.

Fig.1.: the most characteristic peak of graphite (002) is at around 2 theta 25,5 degree. In fig.1. this peak appears at 30 degrees. The same situation is true to the (10) reflection, which appears at around 2 theta 50 degrees instead of 45. Why?  

Site 3, row 105, authors said: “…diffractometer using Co-K-alpha radiation”. in my opinion it was just a hit and the author wanted to write Cu-K-alpha instead of Co-K-alpha.

Author Response

Responses to the Reviewers’ comments (manuscript ID materials-896124)

We would like to thank the Editor and all Reviewers for the effort and time they invested to process and improve our manuscript. We carefully corrected the manuscript according to all of the given suggestions and recommendation. For convenience, the most important changes are marked up in the revised article.

Reviewer 3

Why is there such a big difference between the CNS compositions of ferrous and non-ferrous samples? For example in the case of nitrogen, the N-content in sample CN-Fe is 1.02%, while in the case of CN it is 6.66%, which is more than six times bigger. Or in the case of the Sulphur content: CNS-Fe 1.54% but CNS 5.31%, which is five times bigger. This phenomenon must be explained in the 3.1 session.

RESPONSE: Reviewer 1 raised similar concerns, and as instructed we added appropriate explanation in the manuscript. The differences are due to high-temperature graphitization of carbon caused by its dissolution-precipitation in the iron phase. CN sample does not contain iron and does not undergo catalytic graphitization and for this reason contains large amount of N. However, dissolution of carbon in iron causes its recrystallization and expulsion of heteroatoms and for this reason CN-Fe contains much less N. Iron can dissolve carbon and hence it acts as a good graphitization catalyst. Chars and organic polymers doped with iron undergo extensive graphitization during high-temperature annealing (conversion of amorphous/microporous carbon into crystalline, non-porous graphitic structure). Graphitization of carbon via dissolution of amorphous and precipitation of crystalline carbon over Fe phase causes removal of heteroatoms from carbonaceous phase (e.g. O, H, N) producing pure graphite. However, if sulfur is present in carbonaceous material it reacts instantaneously with iron yielding iron sulfide. Carbon does not dissolve in iron sulfide and hence cannot undergo catalytic graphitization. For this reason, S-, N- and Fe-co-doped amorphous carbon is not converted into graphite and heteroatoms stay in the carbon scaffold. Consequently, after acid leaching of iron mineral phases (sulfides) one obtains amorphous, microporous carbon rich in nitrogen. The reason that CNS-Fe contains only 1.54% but CNS 5.31% of S is that the sulfur is captured by Fe to create FeS and then it is removed during acid leaching of FeS. We have explained this phenomenon in detail in our previous work, which we cite in the current manuscript for reference (Kiciński, W.; Dembinska, B.; Norek, M.; Budner, B.; Polański, M.; Kulesza, P.J.; Dyjak, S. Heterogeneous iron-containing carbon gels as catalysts for oxygen electroreduction: Multifunctional role of sulfur in the formation of efficient systems. Carbon 2017, 116, 655–669, doi: 10.1016/j.carbon.2017.02.045).  

Fig.1.: the most characteristic peak of graphite (002) is at around 2 theta 25,5 degree. In fig.1. this peak appears at 30 degrees. The same situation is true to the (10) reflection, which appears at around 2 theta 50 degrees instead of 45. Why?  

RESPONSE: Indeed, this might be confusing. This is because we have used less common Co-K-alpha instead of more popular Cu-K-alpha radiation. Co lamp is less popular yet more appropriate for collecting XRD spectra of samples containing Fe since in the case of Cu radiation source iron phase gives high fluorescence.

Site 3, row 105, authors said: “…diffractometer using Co-K-alpha radiation”. in my opinion it was just a hit and the author wanted to write Cu-K-alpha instead of Co-K-alpha.

RESPONSE: As we explained above, we did use cobalt (instead of Cu) as an X-ray source.

Reviewer 4 Report

This article investigated the adsorption, catalysis and antibacterial property of carbon-based nano-materials doped with Iron, Nitrogen and Sulphur. The manuscript systematically compared the properties of C-Fe, CS-Fe, CN-Fe, CNS-Fe, C, CS, CN and CNS. The conclusion was based on the detailed characterization and deep analysis of the experimental data. I recommend publication of this article after addressing a few minor issue.

  1. The elemental composition analysis provided the percentages of N, C, H and S. However, the summation of them is not 100%. I suggest the authors indicate what the rest is. Is the rest Fe or O?
  2. In addition to comment 1, there is not error limits value in Table 1. I think each experiment should be repeated several times to insure the reliability of the data. If the authors conducted repeating measurement, please provide the standard deviation of the data.
  3. In the XRD profile, I observed some tiny peaks at around 2θ=63 and 2θ=93 for the samples of CN-Fe and C-Fe. What crystal forms are they corresponding to? Please mark them in the figure.
  4. Why do the author only show the SEM images of Fe-N doped carbon? Was not it better to show all of the samples as a comparison?

Author Response

Responses to the Reviewers’ comments (manuscript ID materials-896124)

We would like to thank the Editor and all Reviewers for the effort and time they invested to process and improve our manuscript. We carefully corrected the manuscript according to all of the given suggestions and recommendation. For convenience, the most important changes are marked up in the revised article.

Reviewer 4

The elemental composition analysis provided the percentages of N, C, H and S. However, the summation of them is not 100%. I suggest the authors indicate what the rest is. Is the rest Fe or O?

RESPONSE: Yes, exactly. Porous carbon materials always contain large amount of surface chemisorbed oxygen (in a wide range of ca. 5 - 25 wt.% depending on the carbon porosity and structure). Usually, the higher the microporosity and the specific surface area are, the higher the amount of oxygen. In addition, the carbon materials can contain some adsorbed water. It is difficult to assess O-content via combustion elemental analysis (additional sophisticated sub-assemblies dedicated to oxygen content measurements are required). We added appropriate explanation in the revised manuscript concerning this particular issue.  

In addition to comment 1, there is not error limits value in Table 1. I think each experiment should be repeated several times to insure the reliability of the data. If the authors conducted repeating measurement, please provide the standard deviation of the data.

RESPONSE: We agree with this important comment. For the elemental analysis three samples were prepared to obtain the final CHNS composition, however each element is detected with slightly different accuracy depending on the nature and initial content of an element (e.g. the accuracy is usually lower for sulfur than for carbon). We made appropriate corrections in Table 1. When it comes to N2 adsorption-desorption isotherms (assessment of SBET) we performed these measurements only once since they are time-consuming (one analysis can last for up to three days), yet they are usually characterized by good reproducibility.

In the XRD profile, I observed some tiny peaks at around 2θ=63 and 2θ=93 for the samples of CN-Fe and C-Fe. What crystal forms are they corresponding to? Please mark them in the figure.

RESPONSE: Similar issue was raised by Reviewer 3. These minor reflections located at higher values of 2-theta can be assigned to graphite, and they indicate significant degree of graphitization of the CN-Fe and C-Fe carbon samples. Please note that here we used diffractometer with Co-K-alpha instead of more common Cu-K-alpha. Consequently, if you compare these spectra with other in literature there is shift in the reflection’s locations into higher values. For details concerning XRD spectra of graphite you can see for instance: X-ray diffraction patterns of graphite and turbostratic carbon, https://doi.org/10.1016/j.carbon.2007.03.038)

Why do the author only show the SEM images of Fe-N doped carbon? Was not it better to show all of the samples as a comparison?

RESPONSE: SEM images of all the studied samples (regardless of the doped elements) are very similar. All the samples are characterized by colloidal morphology, typical for sol-gel derived polymeric resins. Carbon materials obtained from sol-gel synthesized polymers are built of micrometric colloids (spheroids) interconnected into 3D microporous monolithic structure. We described this issue in detail in our previous reports and inspired by this comment we added appropriate explanation with reference (ref. 26) in the revised version of our submission.